# Optical Motion Capture Systems for 3D Kinematic Analysis in Patients with Shoulder Disorders

**DOI:** 10.3390/ijerph191912033

**Published:** 2022-09-23

**Authors:** Umile Giuseppe Longo, Sergio De Salvatore, Arianna Carnevale, Salvatore Maria Tecce, Benedetta Bandini, Alberto Lalli, Emiliano Schena, Vincenzo Denaro

**Affiliations:** 1Research Unit of Orthopaedic and Trauma Surgery, Fondazione Policlinico Universitario Campus Bio-Medico, Via Alvaro del Portillo, 200, 00128 Roma, Italy; 2Research Unit of Orthopaedic and Trauma Surgery, Department of Medicine and Surgery, Università Campus Bio-Medico di Roma, Via Alvaro del Portillo, 21, 00128 Roma, Italy; 3Laboratory of Measurement and Biomedical Instrumentation, Università Campus Bio-Medico di Roma, Via Alvaro del Portillo, 200, 00128 Rome, Italy

**Keywords:** Mocap, motion capture, optical motion capture systems, shoulder, shoulder kinematics, biomechanics

## Abstract

Shoulder dysfunctions represent the third musculoskeletal disorder by frequency. However, monitoring the movement of the shoulder is particularly challenging due to the complexity of the joint kinematics. The 3D kinematic analysis with optical motion capture systems (OMCs) makes it possible to overcome clinical tests’ shortcomings and obtain objective data on the characteristics and quality of movement. This systematic review aims to retrieve the current knowledge about using OMCs for 3D shoulder kinematic analysis in patients with musculoskeletal shoulder disorders and their corresponding clinical relevance. The Preferred Reporting Items for Systematic reviews and Meta-Analyses (PRISMA) guidelines were used to improve the reporting of the review. Studies employing OMCs for 3D kinematic analysis in patients with musculoskeletal shoulder disorders were retrieved. Eleven articles were considered eligible for this study. OMCs can be considered a powerful tool in orthopedic clinical research. The high costs and organizing complexities of experimental setups are likely outweighed by the impact of these systems in guiding clinical practice and patient follow-up. However, additional high-quality studies on using OMCs in clinical practice are required, with standardized protocols and methodologies to make comparing clinical trials easier.

## 1. Introduction

Shoulder dysfunctions (SDY) represent the third musculoskeletal disorder by frequency [1]. Furthermore, SDY negatively impacts the quality of life of affected patients [2,3]. Therefore, SDY is considered a medical and socio-economic problem in Western countries [1,4,5].

To date, the Disabilities of the Arm, Shoulder and Hand (DASH) questionnaire, the Simple Shoulder Test (SST), the Constant Murley score (CMS), the American Shoulder and Elbow Surgeons (ASES) score, and the Visual Analogue Scale (VAS) score for pain and stiffness represent the main tests and scales used in clinical practice for the evaluation of shoulder function [6]. All questionnaires have been shown to have excellent reliability, validity, and responsiveness with a minimal administrative burden [7]. However, despite being applied routinely in clinical practice, each score and scale has a certain degree of subjectivity. Furthermore, these scales do not provide data on joint movement such as speed, deceleration time, or joint range of motion (ROM) [2,8]. Considering the increase in SDY incidence and progressive population ageing, finding new solutions to diagnose and manage this condition could be helpful for clinicians [9,10]. Several authors have attempted to overlap the questionnaire limitations by proposing more detailed alternative assessment scales. For example, in 2016, Cutti et al. [11] proposed modifying the CMS by integrating the original scale with the data obtained from inertial and magnetic measurement units positioned over the thorax, scapula, and humerus to reduce the lack of assessment of scapular dyskinesis.

Shoulder kinematic analysis is a booming research field due to the emerging need to improve clinical diagnostics and rehabilitation procedures [8,12]. From a technological point of view, monitoring the movement of the shoulder is particularly challenging due to the complexity of the joint kinematics, which requires the development of protocols that exploit a detection technology that is as reliable and non-intrusive as possible [8,12]. The kinematic analysis of movement makes it possible to overcome the shortcomings of clinical tests and to obtain objective data on the characteristics and quality of movement [13,14].

The 3D kinematic analysis of the shoulder joint using optical motion capture systems (OMCs) represents a valid instrument for completing the clinical assessment of shoulder functionalities [15]. OMCs are equipped with multiple synchronized cameras estimating the 3D positions of markers by triangulation within a calibrated capture volume [16]. Markers can be passive, i.e., covered by photo-reflective materials that reflect the infrared light, or active, i.e., emitting infrared light [16]. The 3D kinematic analysis of the shoulder joint using OMCs is highly heterogeneous among studies in terms of marker sets, kinematic model, analyzed functional movements, and reported kinematic outcomes [17,18,19]. In kinematic analysis, two marker sets can be used, an anatomical or cluster marker set [14]. The anatomical markers are placed in correspondence with body landmarks to build kinematics models, i.e., anatomical landmarks are essential to define the local coordinate systems of a body segment (e.g., scapula, humerus, and thorax) in static conditions [20]. The main limitations of the anatomical marker sets are related to the movements of the soft tissue underlying the skin, i.e., soft tissue artifact and occlusion during functional movements. The cluster marker sets reduce the aforementioned issues. Clusters of markers with asymmetrical geometrical arrangements must be placed at locations minimizing the effects of soft tissue artifacts and muscular contraction [21].

To the best of our knowledge, no updated systematic review focused on the application of OMCs for the kinematic analysis of patients with musculoskeletal shoulder disorders is available in the literature. This systematic review aims to retrieve the current knowledge about the use of OMCs for 3D shoulder kinematic analysis in patients with SDY and its corresponding clinical relevance.

## 2. Material and Methods

### 2.1. Eligibility Criteria

The research question was formulated using a PIOS approach—Patient (P), Intervention (I), Outcome (O), and Study design (S). Articles describing OMCs (I) applications for 3D shoulder joint kinematic analysis in patients with musculoskeletal shoulder disorders or healthy participants (P) as control groups were selected. The assessed outcomes (O) were the clinical question, tasks included in the assessment protocol, degrees of freedom, kinematic outcomes, and main conclusions. For this purpose, Randomized Controlled Trials (RCT) and Non-Randomized Controlled Trials (NRCTs) as observational studies such as case series (CAS), cross-sectional (CRS) and cohort studies (CS) were considered eligible for the following review (S).

Studies were considered eligible for this review if all of the following inclusion criteria were met: (1) articles published in English; (2) peer-reviewed articles of each level of evidence according to the Oxford classification; (3) articles focusing on using OMCs in the context of shoulder joint disease and task assessment; (4) patients with degenerative shoulder disease, traumatic and atraumatic Rotator Cuff Tear Arthropathy (RCTA), Shoulder Anterior Instability (SAI), scapular dyskinesis, and Frozen Shoulder (FS); and (5) patients who underwent Reverse Total Shoulder Arthroplasty (RTSA), Anatomic Total Shoulder Arthroplasty (aTSA), or arthroscopic rotator cuff repair were included.

Studies were considered ineligible if at least one of the following exclusion criteria was met: (1) technical notes, letters to editors, instructional courses, reviews, books, conference papers, and systematic reviews; (2) in vitro, animal, or cadaveric studies; (3) articles involving only healthy cohorts of volunteers; (4) studies focusing on upper-limb 3D kinematic analysis but excluding the shoulder joint; (5) studies reporting outcomes for patients with pathologies not directly related to shoulder musculoskeletal disorders (e.g., patients undergoing rehabilitation after strokes); (6) studies not reporting the type of OMCs used and/or the tasks included in the assessment protocol; and (7) patients with rheumatoid arthritis or shoulder fracture.

### 2.2. Search Strategy and Studies Selection Process

The Preferred Reporting Items for Systematic reviews and Meta-Analyses (PRISMA) guidelines were used to improve the reporting of the review [22].

The review was carried out by two independent authors (S.M.T. and S.D.S.), considering the available literature from the databases’ inception to July 2022. Differences were reconciled by mutual agreement, and, in case of disagreement regarding the inclusion/exclusion of articles, the consensus of a third reviewer (U.G.L.) was asked. The following keywords were searched in Pubmed, IEEE Xplore, Cochrane, and Web of Science databases: Mocap, “Optical motion capture systems”, and “shoulder kinematics analysis”. The search strategy implemented in Pubmed was: (((Shoulder OR “shoulder joint” OR “shoulder disease*” OR “musculoskeletal shoulder disorder*” OR “shoulder injury*”) AND (rehabilitation OR “physical therapy” OR surgery)) AND (“optical motion capture system” OR “motion capture system” OR Mocap)) AND (“joint angle*” OR “functional task*” OR “range of motion” OR ROM OR kinematic* OR “kinematic outcome*”).

Studies in the reference list of the selected articles were also evaluated to identify any studies omitted in the databases search. Each article was screened first by evaluating the relevance of the title and the abstract. Therefore, the full paper was read for the articles deemed to be included by both reviewers. The number of articles evaluated, excluded, and included in the systematic review was registered and reported in the PRISMA flowchart (Figure 1). Rules by Page et al. were followed in designing the PRISMA chart [22].

### 2.3. Data Synthesis

General study characteristics extracted were primary author, year of publication, type of study, level of evidence (LOE), population demographics, sample size, gender, and mean age (Table 1). Additionally, the OMCs (brand, number of cameras, and sampling frequency), marker positions, and tasks included in the assessment protocol were extracted (Table 2). Furthermore, the aim of the study, the degrees of freedom, the kinematic outcomes, and the main findings were also extrapolated and reported in Table 3.

### 2.4. Quality Assessment

The Risk of Bias (RoB 2) tool for randomized trials, the Risk of Bias in Non-Randomized Studies of Interventions (ROBINS-I) tool by Cochrane, and the Joanna Briggs Institute Critical Appraisal Tool for case series were used to assess the quality of each study [22,32,33]. Each reviewer independently rated selected articles (S.M.T. and S.D.S.). A third reviewer judged in case of disagreement (U.G.L.).

### 2.5. Statistical Analysis

Categorical data were summarized as frequencies and percentages. Continuous data were summarized as mean values, with standard deviations (SD) or range (i.e., minimum, and maximum values). A meta-analysis was not performed at the end of the review due to the heterogeneity of the data of the selected articles.

### 2.6. Systematic Review Registration Statement

The Open Science network Framework was used for the registration statement. The associated doi is https://doi.org/10.17605/OSF.IO/246T9.

## 3. Results

### 3.1. Search Results

The literature search identified 265 studies. No additional studies were found in the gray literature, and no unpublished studies were retrieved. Duplicated article removal excluded 100 papers. Of the remaining 165 articles, 150 were removed as incompatible with the main aim of this review after the title and abstract evaluation. The excluded records were: systematic reviews and technical notes (*n* = 16), in vitro, animal, and cadaveric studies (*n* = 18), healthy cohorts (*n* = 27), shoulder joint not evaluated (*n* = 29), studies not reporting OMCs (*n* = 42), and rheumatoid arthritis or shoulder fracture (*n* = 18). Fifteen full-text articles were then screened, leading to the elimination of four studies. At the end of the selection process, a total of 11 articles were considered eligible for this study [17,18,19,23,24,25,26,27,28,29,30]. The PRISMA flowchart of the literature search is reported in Figure 1.

### 3.2. Study Characteristics and Population

Among the 11 included studies, 6 were cohort studies (CS) of III LOE [17,23,25,27,29,30], 1 was a case series (CAS) of III LOE [18], 2 were retrospective cohort (RCS) studies of III LOE [19,26], 1 was a cross sectional study (CRS) of IV LOE [28], and 1 was a Prospective Laboratory Study (PLS) of III LOE [24]. No RCTs were included in this review. The total number of patients included in the studies was 201. Patients presented the following conditions: glenohumeral osteoarthritis (GOA) in 33 patients [17,18], SAI in 12 patients [24], scapular dyskinesis in 51 patients [28], and RCTA in 34 patients [27,30]. Moreover, studies included 26 patients who underwent aTSA [17,23,26] and 47 patients who underwent RTSA [19,25,29]. Five studies included a healthy control group [17,23,26,27,30], for a total of 55 healthy subjects to compare 3D shoulder kinematics with patients with shoulder musculoskeletal disorders.

### 3.3. Quality Assessment

The ROBINS-I tool for NRCTs and the Joanna Briggs Institute Critical Appraisal Tool for CAS were used to assess the methodological quality of each article [22,32,33]. No RCTs were included in the review. RCCs were identified as low risk of bias studies [28,30] or moderate risk of bias studies [17,23,27]; RCS were overall of good quality [18,19,24,25,29]. The risk of bias assessments for RCTs, NRCTs, and CSs is reported in Figure 2 and Figure 3.

### 3.4. Experimental Setup and Protocols

The OMCs used in the included studies were equipped with several cameras ranging from 6 to 12. Only two studies did not report the number of cameras used in the experimental setup [28,29]. The recording frequency of kinematic data was set to 60 Hz in one study [27], to 100 Hz in two studies [24,29], 120 Hz in four studies [18,19,23,26], to 200 Hz in two studies [25,30], and 240 Hz only in one study [28].

Of all included studies, seven used both anatomical and cluster marker sets [17,18,23,24,28,29,30], and four studies used only the anatomical marker set [19,25,26,27]. Except for two studies [25,26], the body segments of the humerus, thorax, clavicle, forearm, and scapula were defined by following the conventions recommended by the International Society of Biomechanics (ISB) [34].

The main investigated functional upper-limb tasks mainly involved arm elevation in single-plane movements (i.e., sagittal, frontal, scapular, and transverse planes), except for two studies that included tasks mimicking common ADLs (e.g., combing hair, taking a book from a shelf, washing the opposite armpit, and tying an apron) [17,26].

### 3.5. Degrees of Freedom and Kinematic Outcomes

The studies included in this review appropriately described the investigated shoulder degrees of freedom and kinematic outcomes needed to answer the main clinical questions. For each segment (e.g., thorax, clavicle, scapula, humerus, and thorax), the kinematics have been described in terms of Cardan or Euler angles between the distal and proximal segments. As reported in Table 3, for all investigated upper-arm movements, the most analyzed degrees of freedom were the elevations of the humerus relative to the thorax (humerothoracic) and relative to the scapula (glenohumeral) [17,23,26,28]. Shoulder kinematics result from the combined contribution of the glenohumeral joint and the shoulder girdle, including the acromioclavicular and the sternoclavicular joint and the functional scapulothoracic joint [17,23,27]. As scapular kinematic is altered in the presence of musculoskeletal shoulder disorders; several studies evaluated scapular pro-/re-traction, lateral/medial rotation, and posterior/anterior tilting as a function of humeral elevation angles [18,24,27,28,29,30]. Kinematic outcomes were reported in terms of ROM, i.e., the difference between the maximum and minimum elevation angles [17,23], kinematic time series, peak angles, joint angular displacements, or scapulohumeral rhythm (SHR).

## 4. Discussion

The present study aimed to retrieve previous works on 3D shoulder kinematic evaluations in patients with musculoskeletal shoulder disorders and highlight the main outcomes that could have clinical relevance. OMCs represent a powerful tool in the orthopedic clinical practice, mainly when referring to complex joint movement analysis, such as the shoulder joint [35]. These systems allow investigating the joint kinematics in a precise and accurate manner, obtaining quantitative data on joints kinematics [36]. The careful evaluation of postsurgical joint kinematics could allow assessing surgery outcomes to tailor care according to the patient’s characteristics, optimizing postsurgical recovery. In addition, OMCs can be used to evaluate the effectiveness of an ongoing rehabilitation protocol. Therefore, the main information from each of the included studies was extracted to obtain a descriptive synthesis of previous works and bring out their applicability and the transferability of the methodologies and results in common clinical practice.

### 4.1. Markers Setup, Modelling, and Study Protocols

When evaluating residual shoulder functionalities in patients with SDY by using OMCs, it is critical to implement experimental procedures that are as reproducible and impactful as possible for the clinical question underlying the investigation. Quantifying angular joint kinematics of the shoulder joint via OMCs is a relevant method for evaluating the clinical condition of the shoulder in orthopedic patients. However, the methodology and experimental procedures often involve inherent complexity related to the complexity of the shoulder biomechanics and the lack of definitive and standardized methods.

Markers’ configurations and locations are based on the necessity to define a biomechanical model of the shoulder joint and track joint kinematics during dynamic movements in a reliable manner. Among the included studies, body segments that were always included to describe shoulder functionalities were the humerus and the thorax for evaluating the angles of elevation of the humerus relative to the thorax (humerothoracic joint). The thorax segment is defined by placing markers on the following anatomical markers: incisura jugularis (IJ), processus xiphoideus (PX), processus spinosus of the 7th cervical vertebra (C7), and processus spinosus of the 8th vertebra (T8) [18,19,24,29]. The latter are the anatomical landmarks recommended by ISB [34], of which T8 is replaced in some studies with the processus spinosus of the 10th vertebra (T10) [17,23,25,26,28]. The humerus segment is defined by the glenohumeral rotation center and the lateral and medial epicondyles [34]. Since the glenohumeral rotation center is not a palpable anatomical landmark, it should be estimated by regression analysis or recordings of functional movements [34]. Mainly, ISB recommends two techniques for defining the center of rotation of the shoulder joint, i.e., a regression model such as the one proposed by Meskers [37], or the computation of the optimal pivot point of the position vectors of the instantaneous helical axes during motion recordings [34,38]. Of the studies included in this analysis, not all have specified how the center of rotation of the shoulder was determined. Some studies dynamically defined the functional center of rotation, requiring the execution of active movements in all planes of the 3D space [18,26,27,29], which are not always performed optimally by patients with significant limitations of the shoulder. Following the linear regression method proposed by Meskers [37], the glenohumeral joint center is computed by taking into account the anatomical positions of the acromioclavicular (AC) joint, the processus coracoideus (PC), trigonum spinae scapulae (TS), angulus inferior (AI), and angulus acromialis (AA) [19,24].

Kinematic models are defined using anatomical marker sets to determine the local coordinate segments of each segment [14,20]. Kinematic models may also include clusters of markers positioned over the body segment of interest to be tracked during functional tasks and overcome some limitations of purely anatomical models, such as marker occlusion and soft tissue artifacts [14,39]. During static calibration procedures, the relation between markers on the clusters and anatomical landmarks is defined [20]. Of the studies included in this analysis that used a hybrid anatomical–cluster model, clusters of four markers were positioned on the humerus as specified in [17,18,23,24,28]. Biomechanical models of the shoulder joint may also include the clavicle and scapula segments [17,18,23,27]. Robert-Lachaine et al. [27] analyzed the kinematics of the shoulder, including anatomical markers and technical markers positioned over body regions of interest which minimize the movements of the underlying soft tissue. In particular, the clavicle segment was defined by five markers (including both anatomical and technical ones), and the scapula segment was defined by nine markers [27].

The ability to monitor scapular kinematics is valuable in managing patients with abnormal scapular motion patterns due to pathologic conditions. Probably, the scapular segment is the most challenging to track because it does not have a fixed center of rotation, it has a particular morphological configuration and sliding on the thorax, and the variations in its orientation during upper-limb movements result in conspicuous deformations of the surrounding soft tissues. Of course, the gold standard for scapular movement tracking is the method using bone pin insertion [40,41]. Nevertheless, this method is not feasible for implementation during common clinical practice because it is invasive and requires anesthetizing. Over the years, significantly less invasive methods have been proposed for functional assessment of the scapula using OMCs. These include the acromion cluster method, in which a cluster of three or four markers is generally placed over the flat portion of the acromion, then calibrated with the anatomical coordinate system of the scapula defined by the TA, AI, and AA [42,43]. Seven of the studies included in this paper used an acromion marker cluster (AMC) to track scapular kinematics during upper-limb movements [17,18,23,24,28,29,30]. Although this method has been investigated and validated up to 120° of humeral elevation, care must be taken in interpreting the results above this elevation angle [40,44].

When aiming to assess the functional level of the shoulder, it is important to select movements that have clinical significance and that reproducibly investigate relevant aspects of the shoulder joint oriented toward achieving the predefined clinical questions [14]. Most movement protocols selected from the studies included in this paper were goal-driven and could be split into two main categories: standardized movements performed in known planes of the 3D space and more complex movements that simulate activities of daily living. Elevations in the sagittal plane were evaluated in three of the included studies [18,19,23], elevations in the frontal plane were evaluated in two studies [23,24], and elevations in the scapular plane (defined as 40° anteriorly rotated with respect to the frontal plane [19,27]) were evaluated in five studies [19,25,27,29,30]. Only one study required patients with traumatic anterior instability to make intra-/extra-rotation movements in the transverse plane while keeping the arm adducted and the elbow flexed at 90° [24]. Patients with GOA and who received aTSA performed several activities of daily living during the study protocol, such as perineal care, washing axilla, combing hair, taking a book from a shelf, and tying an apron [17,26]. Ueda et al. [28], who investigated the association of scapular kinematics alteration and glenohumeral joint with the type of scapular dyskinesis, asked baseball players to perform pitching motions specific to their sports category. Therefore, the analysis of shoulder kinematics must be based on the choice of movement protocols aimed at achieving the clinical goal, although much heterogeneity is present among studies in the literature. Moreover, given the complexity of shoulder biomechanics and the wide ROM that this joint is able to perform, the selection of movements to be investigated is often limited to a specific category or a limited number of actions that could have high relevance when related to common activities of daily living.

### 4.2. Degrees of Freedom and Kinematic Outcomes

For describing the motion of the humerus relative to the thorax (humerothoracic joint) and of the humerus relative to the scapula (glenohumeral joint), ISB recommends the YXY Euler sequence [34]. Of the eligible studies investigated in this study, several studies used the YXY Euler sequence to compute the humerothoracic and glenohumeral joints degrees of freedom [18,27,28,29]. However, previous studies have shown the occurrence of kinematic singularities at 0° and 180° of humeral elevation in the frontal and sagittal planes [45,46], a phenomenon known as gimbal lock. This last one is described as the loss of one degree of freedom occurring when two axes become parallel during motions [45]. For this reason, alternative rotation sequences have been proposed. The YXY Euler rotation sequence has been compared with the Cardan rotation sequence XZY to describe the glenohumeral movement in the scapular plane [45,47]. Results show that the YXY rotation sequence described the humerus in a more externally and anteriorly rotated position compared with the XZY sequence [47]. The authors also suggested using the XZY rotation sequence for a better clinical interpretation of the kinematic results [47]. Other studies confirmed these findings and suggested the sequence of flexion–extension, abduction–adduction, and internal–external rotation to describe the motion of the humerus relative to the thorax during movements performed in the sagittal plane, and the sequence of abduction–adduction, flexion–extension, and internal–external rotation to describe the motion of the humerus relative to the thorax during movements in the frontal plane [48,49], as specified in the study of Dellabiancia et al. [24]. The rotation sequence YXZ was used to evaluate the kinematics of the scapula relative to the thorax (scapulothoracic joint) [18,19,27,29], following the ISB recommendations [34]. Movements around the Y-, X-, and Z-axes represent internal–external rotation, upward–downward rotation, and anterior–posterior tilting, respectively [27,34].

The studies included in this paper investigated several kinematic outcomes. In particular, among the studies, the kinematic outcomes’ graphical representation varied from kinematic time series [17] to joint angles (°) or joint angles displacements (∆°) vs. angles (°) [25,29]. Moreover, numerical kinematic outcomes were provided. Of these, the most widely reported variables were the maximum and minimum angular values, peak angles as points of task achievement, and ROM defined as the difference between the maximum and minimum angular values reached during each repetition of a task [17,19,24,25,27,28,30,34]. All alterations in scapular orientation and in the control afforded by the stabilizing muscles of the scapula are believed to disrupt the stability and function of the glenohumeral joint, thus contributing to SDY, such as shoulder impingement, rotator cuff pathology, or shoulder instability [50]. A coordinated scapular kinematic represents a critical component for the correct functionality of the shoulder joint [51]. Therefore, some studies aimed to evaluate the contribution of scapular kinematics in shoulder pathology. While some studies evaluate shoulder kinematics up to predetermined elevation angles, others have required patients to reach their maximum degrees of elevation without feeling pain [25,27,29]. In the study of Lee et al. [19], patients after RTSA performed elevations in the sagittal and frontal plane bilaterally to reduce the compensatory movements of the thorax. Although the mean humeral elevation angles in the sagittal and frontal plane were 125.1° for RTSA shoulders and 143.0° for the contralateral arm [24], the angle-to-angle plots (scapulothoracic angles vs. humeral elevation angles) were reported up to 120° of humeral elevation. This limitation is also reported in other studies because of the known validity of AMC up to 120° [18] and the availability of humeral elevation angles from patients up to 90° because of pathologic conditions [23]. However, other studies have reported the scapulothoracic angular displacements (∆°) and scapulothoracic angles (°) vs. the humerothoracic elevation angle (°) reaching elevation angles greater than 120° [25,29]. Arm elevation is the most extensively studied shoulder function to figure out the contribution and relationship between the glenohumeral and scapulothoracic joints, namely, the so-called SHR [52]. The overall glenohumeral-to-scapulothoracic motion ratio is 2:1 [23,52]. Several of the studies included in this paper evaluate the SHR, although there is significant variability in its computation. Bruttel et al. [23] calculated the SHR as the ratio between the mean glenohumeral contribution and the mean contribution of the shoulder girdle, which includes the sternoclavicular and acromioclavicular joint and the scapulothoracic joint. Similarly, in the studies conducted by Spranz et al. [18] and Robert-Lachaine et al. [27], the SHR was evaluated using a 3D method in which all three angles of a joint (and not a selected angle) are considered to account for the overall contribution to arm elevation with respect to the thorax [27,31]. In other studies, the SHR was evaluated as the ratio of the glenohumeral elevation to scapulohumeral upward rotation [19,25,29,30]. In particular, the SHR was calculated as the ratio of the difference between humerothoracic elevation and scapulothoracic upward rotation to scapular upward rotation [19,29].

New kinematic parameters could be investigated in future studies to evaluate the overall effect of an intervention (surgical or conservative) in patients with SDY, although this would require greater standardization in both experimental procedures and kinematic analysis to facilitate better comparisons between studies evaluating the same patient population.

### 4.3. Clinical Relevance and Limitations of 3D Kinematic Analysis of the Shoulder with OMCs

Objective measurements of the shoulder joint kinematics have a crucial role in understanding movement disorders or assessing the outcomes of a particular surgical technique or rehabilitation process. Although other sensing elements or technology such as magneto-inertial measurement units, strain sensors, and virtual reality are emerging solutions for motion tracking in an unstructured environment [8,12,53], OMCs employing body markers are currently considered the gold standard in research and clinical applications [8,14]. According to the retrieved studies, the 3D shoulder kinematic analysis with OMCs is a core part of evaluating treatment outcomes and shoulder functionalities during the recovery process of patients. However, among the reported studies, only one performed a 3-year longitudinal study to evaluate motion patterns during activities of daily living in patients scheduled for total shoulder arthroplasty before surgery, at 6 months, and 3 years after shoulder replacement. The simultaneous application of objective kinematics analysis by employing OMCs and clinical scales at different follow-ups may be helpful for clinicians, since a complete clinical picture could be provided for each patient. Moreover, the integration of OMCs in common clinical practice is strengthened by the possibility of objectively assessing compensatory movements that otherwise would not be evidenced. Although none of the included studies assess the thorax degrees of freedom, the inclusion of thorax movements with respect to the global reference system could delineate strategies needed to restore proper shoulder elevation without compensatory movements (such as thorax tilt). The main limitations of 3D kinematic analysis of the shoulder with OMCs can be referred to soft tissue artifacts and marker occlusions that, although they are well-known in general for OMCs, could be more evident during the study of complex movements of the shoulder joint. For this reason, much care must be devoted to the placement of markers at anatomical landmarks and study protocol. Moreover, as revealed by the included studies, still for the upper limb, there are non-standardized kinematic models in the literature, i.e., markers configuration, static calibration postures (not always made explicit in the studies’ methodology) or functional movements for determining joints centers of rotation.

## 5. Conclusions

OMCs represent a powerful tool in orthopedic clinical research for the management of patients with SDY. The clinical implications regarding OMCs are several: OMCs allow for the objective evaluation of shoulder joint kinematics, which would otherwise be biased by operator subjectivity. Thus, within the clinical setup, OMCs represent a significant aiding tool for the surgical team to rely upon, as objective preoperative ROM measurements should become a key factor influencing surgical plans. Moreover, the improvement in their accuracy and reliability should make postoperative ROM central in postsurgical evaluations and follow-up.

Even though these systems have not always proved to be suitable in a clinical setting due to their expensiveness and postprocessing analysis, it has been recently shown that they can be more easily used as a reliable measurement tool for shoulder kinematics [36]. The high costs and complexities of experimental setups are outweighed by the impact of these systems in guiding clinical practice and patient follow-up. However, additional high-quality studies on using OMCs for shoulder kinematics analysis in clinical practice are required, with standardized protocol homogeneity among studies.

## Figures and Tables

**Figure 1 ijerph-19-12033-f001:**
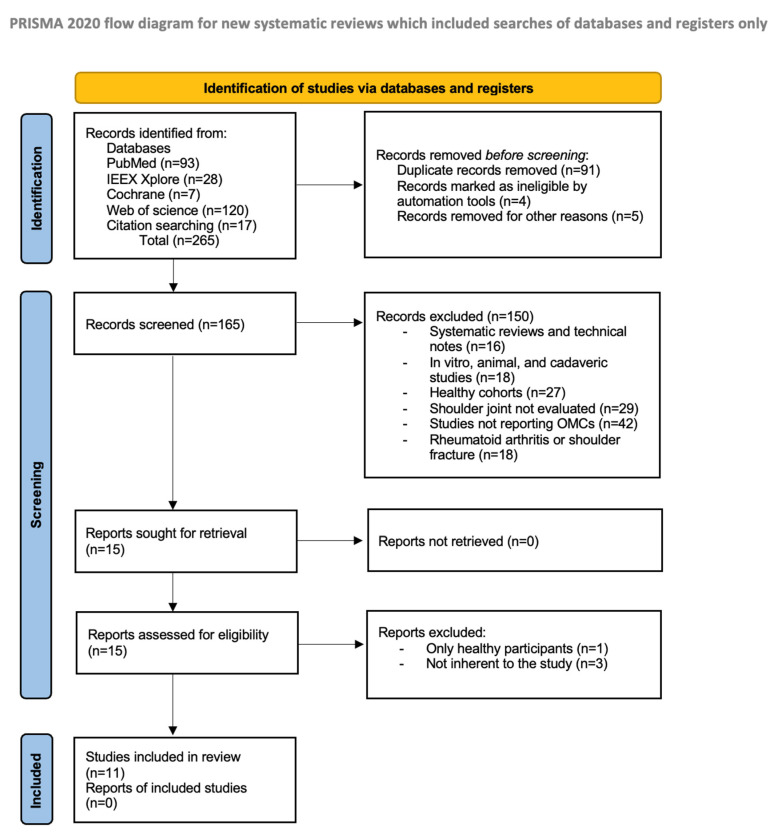
Study selection process and screening according to the PRISMA flow chart.

**Figure 2 ijerph-19-12033-f002:**
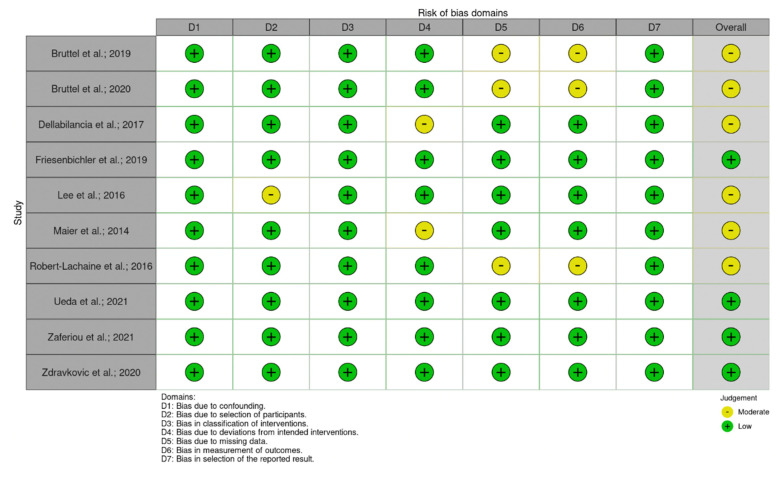
The risk of bias assessments for NRCTs studies according to ROBINS-I tool [17,19,23,24,25,26,27,28,29,30].

**Figure 3 ijerph-19-12033-f003:**
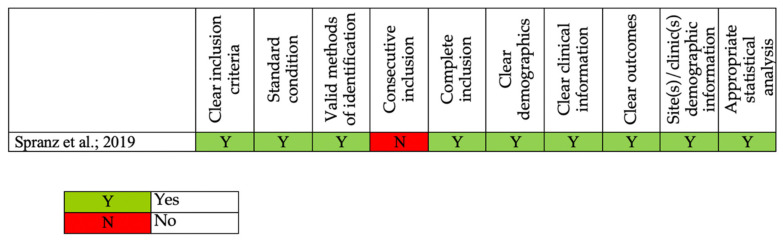
The risk of bias assessments according to Joanna Briggs Institute Critical Appraisal Tool for Case Series [18].

**Table 1 ijerph-19-12033-t001:** Population demographics.

Author and Year	Type of Study	LOE	Population			
			Patients	Gender		Age (Mean ± SD or Mean (Range))
				M	F	
Bruttel et al., 2019 [17]	CS	III	HC: 11	4	7	69.6 ± 5.3
			aTSA: 16 *	7	9	71.2 ± 5.2
			GOA: 12 *	8	4	66.3 ± 7.5
Bruttel et al., 2020 [23]	CS	III	HC: 11 **	4	7	69.6 ± 5.3
			aTSA: 16 **	7	9	71.2 ± 5.2
Dellabiancia et al., 2017 [24]	PLS involving Human subjects	III	SAI: 12	8	4	26 (18–32)
Friesenbichler et al., 2019 [25]	CS	III	RTSA: 23	11	12	74 ± 7
Lee et al., 2016 [19]	RCS	III	RTSA: 13	1	12	72 (69–79)
Maier et al., 2014 [26]	RCS	III	HC: 10	5	5	64.0 ± 7.3
			aTSA (GOA): 10	3	7	65.0 ± 4.7
Robert-Lachaine et al., 2016 [27]	CS	III	HC: 14	12	2	25.2 ± 4.1
			RCTA: 14	12	2	56.4 ± 6.3
Spranz et al., 2019 [18]	CAS	III	GOA: 21	13	8	64.3 ± 9.2 (46–72)
Ueda et al., 2021 [28]	CRS	IV	TI-SD: 24	24	-	20 ± 1
			TIV-SD: 27	27	-	20 ± 1
Zaferiou et al., 2021 [29]	CS	III	RTSA: 11	6	5	71 ± 7
Zdravkovic et al., 2020 [30]	CS	III	HC: 20	13	7	27 ± 3.5
			RCTA: 20	10	10	74 ± 6.2

LOE: level of evidence; CS: cohort study; CRS: cross-sectional study; CAS: case-series; RCS: Retrospective Cohort Study; HC healthy control; PLS: Prospective Laboratory Study; RTSA: Reverse Total Shoulder Arthroplasty; GOA: glenohumeral osteoarthritis; aTSA: Anatomic Total Shoulder Arthroplasty; SAI: Shoulder Anterior Instability; RCTA: Rotator Cuff Tear Arthropathy; TI-SD: type I scapular dyskinesis; TIV-SD: type IV scapular dyskinesis; M: male; F: female; SD: standard deviation. * Two patients were included in both aTSA and GOA groups. ** The aTSA and HC groups in [17,23] are the same, so, they were counted once.

**Table 2 ijerph-19-12033-t002:** Study settings and movement protocols.

Author and Year	OMCs (Brand, Number of Cameras, Sampling Frequency	Marker Set (Number of Markers)	Markers Position	Tasks
Bruttel et al., 2019 [17]	Vicon 12 NR	Anatomical Cluster	Clavicle, forearm, humerus, scapula, thorax B, **	Perineal care Washing axilla Combing hair Taking a book from a shelf
Bruttel et al., 2020 [23]	Vicon 12 120 Hz	Anatomical Cluster (4 markers for humerus and scapular clusters)	C7, T10, IJ, PX, acromion cluster, humerus cluster, digitized anatomical landmarks B, **	E, sagittal plane E, frontal plane
Dellabiancia et al., 2017 [24]	Vicon 8 100 Hz	Anatomical (20) Cluster (4 markers for humerus and scapular clusters; 3 markers for thorax and scapular clusters)	C7, T8, IJ, PX, TS, AI, AA, PC, LE, ME, RS, US 4 markers cluster on humerus and forearm 3 markers cluster on scapula and thorax B, **	AB-AD, frontal plane IR-ER, transverse plane (arm adducted, elbow 90° of flexion)
Friesenbichler et al.; 2019 [25]	Vicon 12 200 Hz	NR	C7, T10, IJ, PX, AI, US, RS, U, TD B	E, scapular plane
Lee et al., 2016 [19]	Motion Analysis Co. 6 120 Hz	Anatomical (16)	TS, AI, AA, midpoint between the most anterosuperior aspect of the acromioclavicular joint and the angle of the acromion, C7, T8, IJ, PX, LE, ME B, **	E, sagittal plane E, scapular plane
Maier et al., 2014 [26]	Vicon 312 12 120 Hz	Anatomical (14)	IJ, PX, C7, T10, AC, ulna distally to the olecranon, RS, US, tuberositas deltoidea	Combing the hair Washing the opposite armpit Tying an apron Taking a book from a shelf
Robert-Lachaine et al., 2016 [31]	Vicon 8 60 Hz	Anatomical (35)	Pelvis (4), trunk (6), clavicle (5), scapula (9), upper arm (7), lower arm (4) B, **	E, scapular plane
Spranz et al., 2019 [18]	Vicon 12 120 Hz	Anatomical (18) Cluster (4 markers for humerus and scapular clusters)	C7, T8, IJ, PX, TS, AI, AA, AC, LE, ME, SC, Acromion clusters, humerus clusters B, **	E, sagittal plane E, frontal plane
Ueda et al., 2021 [28]	MAC3D system NR 240 Hz	Anatomical Cluster (4 markers for scapular clusters)	C7, T10, IJ, PX, LE, ME, acromion clusters, hand B, **	Pitching motion
Zaferiou et al., 2021 [29]	Optitrack system NR 100 Hz	Anatomical Cluster (3 markers for scapular clusters)	C7, T8, IJ, PX, LE, ME, AC, acromion clusters B, **	E, scapular plane
Zdravkovic et al., 2020 [30]	Vicon 8 200 Hz	Anatomical Cluster (25)	Trunk, arms, shoulders, B, **	E, scapular plane

OMC: optical motion capture; NR: non reported; B: bilateral; **: Recommendations of the International Society of Biomechanics; C7: processus spinosus of the 7th cervical vertebra; T8: processus spinosus of the 8th thoracic vertebra; T10: processus spinosus of the 10th thoracic vertebra; IJ: incisura jugularis; PX: processus xiphoideus; TS: trigonum spinae scapulae; AI: angulus inferior of the scapula; AA: angulus acromialis; PC: processus coracoideus; LE: lateral epicondyle; ME: medial epicondyle; RS: radial styloid; US: ulnar styloid; U: ulna distally to the olecranon; TD: tuberositas deltoidei; AC: acromioclavicular junction; SC: sternoclavicular junction; E: elevation; AB: abduction; AD: adduction; IR: internal rotation; ER: external rotation.

**Table 3 ijerph-19-12033-t003:** Aim, shoulder kinematic outcomes, and main conclusions.

Author and Year	Aim	Degrees of Freedom	Kinematic Outcomes	Conclusions
Bruttel et al., 2019 [17]	To examine how aTSA improves the performance in daily activities compared with patients with GOA and healthy controls	Humerothoracic elevation Glenohumeral elevation	ROM Peak angles Kinematic time series	Total shoulder arthroplasty improves the performance of activities of daily living in patients with primary GOA but cannot restore the full ROM compared with healthy controls.
Bruttel et al., 2020 [23]	To confirm a higher amount of scapula lateral rotation to compensate for reduced glenohumeral elevation after aTSA and examine additional effects on the sternoclavicular and acromioclavicular joints’ kinematics	Sternoclavicular pro-/retraction Sternoclavicular elevation/depression Glenohumeral elevation/depression Acromioclavicular pro-/retraction Acromioclavicular lateral/medial rotation Acromioclavicular posterior/anterior tilting Humerothoracic elevation	cSG = AUC (SG elevation)/AUC (HT elevation) cGH = 1 − cSG SHR = (1 − cSG)/cSG	The SG relative contribution to the elevation movements in patients after aTSA is higher than in healthy controls. Kinematics of sternoclavicular and acromioclavicular joints showed significantly different patterns.
Dellabiancia et al., 2017 [24]	To assess the effectiveness of a novel glenohumeral joint immobilizer	Scapular pro-/retraction Scapular lateral/medial rotation Scapular posterior/anterior tilting Humeral abduction–adduction Humeral internal–external rotation	ROM Glenohumeral translation in superior–inferior direction Euclidean distance between glenohumeral centre of rotation and the geometrical centre (i.e., the trunk).	The immobilizer significantly limited joint excursion in all planes of movement except internal rotation.
Friesenbichler et al., 2019 [25]	To demonstrate the differences in scapulothoracic joint contribution to shoulder abduction in RTSA patients with poor-to-excellent function	Scapular lateral/medial rotation Glenohumeral abduction	ROM Peak angles SHR	Limited shoulder abduction is not associated with insufficient scapulothoracic mobility after RTSA.
Lee et al., 2016 [19]	To evaluate the dynamic 3D scapular motion in addition to the SHR in the RTSA and contralateral shoulders during dynamic arm motion.	Scapular lateral rotation Scapular internal rotation Scapular posterior tilting Humeral elevation	Peak angles SHR	Increased scapular lateral rotation and decreased SHR after RTSA indicate that RTSA shoulders use more scapulothoracic motion and less glenohumeral motion to elevate the arm.
Maier et al., 2014 [26]	To examine whether total shoulder arthroplasty is able to restore normal ROM in ADLs in patients with degenerative GOA over the course of 3 years.	Humerothoracic abduction/adduction Humerothoracic external rotation Humerothoracic flexion/extension	Maximum angles Minimum angles ROM	aTSA improves the ability to perform ADLs in patients with degenerative GOA. However, these patients do not use their maximum available abduction ROM in performing ADLs.
Robert-Lachaine et al., 2016 [27]	To identify the SHR patterns of compensation to reach the maximal arm elevation without pain in patients with symptomatic rotator cuff tears compared with a control healthy group.	Scapular lateral rotation Scapular internal rotation Scapular posterior tilting Humeral elevation	SHR ROM	Patients who reached at least 85° showed reduced SHR as they compensated for the loss of glenohumeral motion by increased scapulothoracic contribution. Patients who reached at least 40° showed increased SHR since they underused the scapulothoracic joint.
Spranz et al., 2019 [18]	To investigate the variation of the glenohumeral and scapulothoracic motion in progressive severity GOA.	Scapulothoracic elevation Humeral elevation	cST = AUC (ST elevation)/AUC (HT elevation) SHR	In the progressive severity of GOA, the contribution of the scapulothoracic joint to the total humeral elevation between 30° and 90° increased to compensate the loss of glenohumeral joint movement.
Ueda et al., 2021 [28]	To clarify the incidence of scapular dyskinesis types in baseball players and investigate kinematic alterations in glenohumeral joint and scapular motion during pitching in baseball players with type I scapular dyskinesis.	Scapular internal rotation Scapular lateral rotation Scapular posterior tilting Glenohumeral horizontal abduction Glenohumeral external rotation Glenohumeral abduction Humerothoracic horizontal abduction Humerothoracic external rotation Humerothoracic abduction	ROM Joint angles Peak values	Baseball players in the abnormal group showed increased glenohumeral motion and decreased scapular motion during pitching compared with the normal group.
Zaferiou et al., 2021 [29]	To compare SHR used before and after RTSA during the ascent phase of scapular plane arm elevation tasks performed with varied shoulder rotation.	Scapular pro-/retraction Scapular lateral/medial rotation Scapular posterior/anterior tilting Humeral elevation	SHR Joint angular displacements	This study showed significant differences in scapulohumeral coordination before vs. after RTSA aligned with the hypothesis of increased SHR post-RTSA.
Zdravkovic et al., 2020 [30]	To evaluate the SHR variations in adults with and without RCA during arm elevation	Scapular pro-/retraction Scapular lateral/medial rotation Scapular posterior/anterior tilting Humeral elevation Humeral flexion	SHR Peak values	Patients with RCA exhibited more scapulothoracic motion during arm elevation than the control group.

aTSA: anatomical total shoulder arthroplasty; GOA: glenohumeral osteoarthritis; SG: shoulder girdle; cSG: mean shoulder girdle contribution; HT: humerothoracic; AUC: area under the curve; cGH: mean glenohumeral contribution; SHR: scapulohumeral rhythm; cST: scapulothoracic contribution; RCA: rotator cuff arthropathy; ROM: range of motion; ADLs: activities of daily living; RTSA: Reverse Total Shoulder Arthroplasty.

## Data Availability

The data presented in this study are available on request from the corresponding author.

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
