# Peer review of "Optical Motion Capture Systems for 3D Kinematic Analysis in Patients with Shoulder Disorders"

_ijerph, 2022, doi:10.3390/ijerph191912033_

Round 1
Reviewer 1 Report
I think this is a solid paper. I do not have any comments beyond the request for more detail for the 150 papers that are excluded in this literature review from both Figure 1 and its accompanying explanation (Page 5 line 159, 160).
Author Response
Dear reviewer, thank you for the helpful comments and suggestions you made, that contributed in improving the overall quality of the paper.
We reviewed the paper accordingly and hope that the work is now ready for publication.
Our comments (dark blue text) follow the reviewer’s suggestions below.
Reviewer’s comments:
I think this is a solid paper. I do not have any comments beyond the request for more detail for the 150 papers that are excluded in this literature review from both Figure 1 and its accompanying explanation (Page 5 line 159, 160).
Thank you for the observation. We reported the updated details that lead to the exclusion of the articles both in the PRISMA flow chart (Figure 1) and in the corresponding paragraph in the text (Page 5, Paragraph 3.1: “Search results”).
Reviewer 2 Report
I think the article is suitable for publication inthe present form. Methods are clear and reproducible, language is adequate. The only criticism to be moved is the presence of a high number of self-citations. I suggest to replace some of them with other articles.
Author Response
Dear reviewer, thank you for the helpful comments and suggestions you made, that contributed in improving the overall quality of the paper.
We reviewed the paper accordingly and hope that the work is now ready for publication.
Our comments (dark blue text) follow the reviewer’s suggestions below.
Reviewer’s comments:
I think the article is suitable for publication in the present form. Methods are clear and reproducible, language is adequate. The only criticism to be moved is the presence of a high number of self-citations. I suggest to replace some of them with other articles.
Thank you for your comment. We proceeded by substituting or changing some of the citations that were highlighted. Referring to the previous manuscript, the following citations were deleted: citation number 4; 6; 13; 16.
Reviewer 3 Report
This is a very thorough but 30 thousand foot assessment of kinematic studies of the shoulder joint that leaves few conclusions and does not allow much comparison of the different methods. Practicing surgeons like to have clear facts that researchers can offer to guide clinical treatments. I see none of that in this paper.
Few will read this paper and even fewer will understand the implications. A very comprehensive body of work though.
Author Response
Dear reviewer, thank you for the helpful comments and suggestions you made, that contributed in improving the overall quality of the paper.
We reviewed the paper accordingly and hope that the work is now ready for publication.
Our comments (dark blue text) follow the reviewer’s suggestions below.
Changes in the text are highlighted in yellow.
Reviewer’s comments:
This is a very thorough but 30 thousand foot assessment of kinematic studies of the shoulder joint that leaves few conclusions and does not allow much comparison of the different methods. Practicing surgeons like to have clear facts that researchers can offer to guide clinical treatments. I see none of that in this paper.
Few will read this paper and even fewer will understand the implications. A very comprehensive body of work though.
Thank you for the observation. We added a paragraph in the conclusion section where we highlighted how, in the current practice, OMCs represent an objective for the measurement of preoperative and postoperative ROM, which overcome the limitations that the current tools provide. They thus should be a tool for the surgical team to rely on in everyday practice, if available in the facility.
“The clinical implications regarding OMCs are several: OMCs allow for the objective evaluation of shoulder joint kinematics, which would be otherwise be biased by the operator subjectivity. Thus, within the clinical setup, OMCs represent a significant aiding tool for the surgical team to rely upon, as objective preoperative ROM measurements should become a key factor influencing the surgical plan. Also, the improvement in their accuracy and reliability should make postoperative ROM central in post-surgical evaluations and follow-up.
Even though these systems have not always proved to be suitable in a clinical setting due to their expensiveness and post processing analysis, it has been recently shown that they can be more easily used as a reliable measurement tool for shoulder kinematics”